# Phenazine production promotes antibiotic tolerance and metabolic heterogeneity in *Pseudomonas aeruginosa* biofilms

Konstanze T. Schiessl[1], Fanghao Hu [2], Jeanyoung Jo[1], Sakila Z. Nazia[1], Bryan Wang[1], Alexa Price-Whelan[1], Wei Min [2] & Lars E.P. Dietrich [1]

Antibiotic efficacy can be antagonized by bioactive metabolites and other drugs present at infection sites. *Pseudomonas aeruginosa*, a common cause of biofilm-based infections, releases metabolites called phenazines that accept electrons to support cellular redox balancing. Here, we find that phenazines promote tolerance to clinically relevant antibiotics, such as ciprofloxacin, in *P. aeruginosa* biofilms and that this effect depends on the carbon source provided for growth. We couple stable isotope labeling with stimulated Raman scattering microscopy to visualize biofilm metabolic activity in situ. This approach shows that phenazines promote metabolism in microaerobic biofilm regions and influence metabolic responses to ciprofloxacin treatment. Consistent with roles of specific respiratory complexes in supporting phenazine utilization in biofilms, phenazine-dependent survival on ciprofloxacin is diminished in mutants lacking these enzymes. Our work introduces a technique for the chemical imaging of biosynthetic activity in biofilms and highlights complex interactions between bacterial products, their effects on biofilm metabolism, and the antibiotics we use to treat infections.

[1] Department of Biological Sciences, Columbia University, New York, NY 10027, USA. [2] Department of Chemistry, Columbia University, New York, NY 10027, USA. These authors contributed equally: Konstanze T. Schiessl, Fanghao Hu. Correspondence and requests for materials should be addressed to W.M. (email: wm2256@columbia.edu) or to L.E.P.D. (email: LDietrich@columbia.edu)

High-throughput screens of drug–drug combinations have revealed that synergistic, antagonistic or suppressive interactions arise frequently[1–3]. Small molecules produced by bacteria similarly have the potential to influence bacterial susceptibility to antibiotic treatment. These compounds, commonly released during the stationary phase of growth in liquid cultures, during growth in biofilms, and during infection, can impact the physiology of their bacterial producers, as well as other microbes or eukaryotic hosts, in line with their numerous functions, such as nutrient acquisition, signaling or inhibition of competitors[4–6]. One of the few well-described examples of a small molecule metabolite that alters antibiotic efficacy is that of indole, a signaling molecule released by *Escherichia coli* and other bacteria[7,8]. Indole antagonizes the effect of antibiotic treatment by inducing expression of efflux pumps and oxidative stress responses.

*Pseudomonas aeruginosa* produces redox-active pigments called phenazines that affect gene expression, metabolic flux, and redox balancing in their producers[9,10] and that have the potential to alter antibiotic susceptibility. *P. aeruginosa* is a major cause of hospital-acquired infections and chronic lung infections in patients with the inherited disease cystic fibrosis. In addition to phenazine production, a salient feature of *P. aeruginosa* infections is the formation of biofilms, densely packed communities with limited oxygen at depth. *P. aeruginosa* has few metabolic strategies to support redox homeostasis under anoxic conditions (including the use of nitrate[11] or, to a limited degree, arginine[12] or pyruvate[13] fermentation). Analyses of biofilm growth and gene expression have indeed indicated that denitrification and pyruvate fermentation occur in biofilms grown under an oxic atmosphere, and that components of these pathways are not uniformly expressed over biofilm depth[10,14,15]. Furthermore, as endogenous phenazines constitute an alternate electron acceptor[16–18], it has also been proposed that they support metabolic activity in hypoxic biofilm subregions[10,16,19]. Consistent with this model, microelectrode measurements of intact biofilms show that the extracellular phenazine pool becomes more reduced at depth[20], suggesting that cells in the oxygen-limited biofilm base carry out phenazine reduction. Studies of metabolic mutants have also implicated *P. aeruginosa*'s $cbb_3$-type terminal oxidases Cco1 and Cco2, important components of the respiratory chain, in this activity[20]. Collectively, these results provide indirect evidence that metabolism is qualitatively heterogeneous over biofilm depth.

Metabolic heterogeneity in biofilms could equate to differences in antibiotic susceptibility and impede treatment of biofilm-based infections. Here, we investigate the effect of *P. aeruginosa* phenazine production on the survival of cells in biofilms that have been exposed to antibiotics. By combining stable isotope labeling and stimulated Raman scattering (SRS) microscopy, we develop a chemical imaging technique to visualize biofilm metabolic activity in situ. We use this technique to assess the influence of phenazine synthesis and antibiotic treatment on metabolism across biofilm depth. Finally, we use metabolic mutants to test whether specific phenazine-fostered pathways contribute to survival during antibiotic exposure. Our results underscore the relevance of endogenous bacterial products to community behavior and potential therapeutic approaches.

## Results

**Phenazine synthesis aids survival in antibiotic-exposed biofilms.** To test the effect of phenazine production on antibiotic treatment, we chose to work with *P. aeruginosa* PA14 colony biofilms grown on a chemically defined medium with glucose as the sole carbon source. Under these conditions, wild-type colonies produced phenazine-1-carboxylic acid (PCA) and phenazine-1-carboxamide (PCN; Fig. 1a, b), visible as yellow coloration, while the methylated phenazines commonly associated with *P. aeruginosa* cultures and infections were not detectable (WT; Supplementary Figure 1). For the following experiments, we quantified the effect of these endogenously produced phenazines on metabolism and antibiotic efficacy by comparing WT and a strain that completely lacks phenazine production (Δphz)[21].

Pre-grown colony biofilms were exposed for 24 h to antibiotics from different classes and then collected, homogenized, and plated for colony-forming units (CFUs) (Fig. 1c). For the remainder of this manuscript, we will use the terms "survival" and "tolerance" to describe the formation of CFUs by *P. aeruginosa* cells from such antibiotic-treated biofilms. We observed broad antagonistic effects: phenazines diminished killing by the aminoglycoside tobramycin, the beta-lactam carbenicillin (Supplementary Figure 2a, b), and the fluoroquinolone ciprofloxacin (Fig. 1d). The polymyxin colistin was the only antibiotic for which phenazines acted synergistically (Supplementary Figure 2c), i.e. increased susceptibility. Notably, the minimum inhibitory concentration (MIC) determined in shaken liquid cultures did not differ between WT and Δphz for any of the antibiotics (Supplementary Figure 2d), nor was there any significant difference in CFU counts between strains for untreated biofilms (Supplementary Figure 3). For further experiments, we focused on the clinically relevant antibiotic ciprofloxacin, because it was the most effective at killing biofilm cells and because it was the antibiotic for which phenazines had the strongest antagonistic effect. Protection by phenazines was not significant when stationary-phase liquid cultures were subjected to ciprofloxacin treatment (Supplementary Figure 4), and though the addition of pure phenazines provided some protection in liquid culture, it was only detectable in a limited range (Supplementary Figure 5). Together, these results show that phenazines antagonize the effects of ciprofloxacin on cells grown in colony biofilms; i.e., phenazine exposure allows more cells from antibiotic-treated biofilms to survive treatment and subsequently form CFUs when plated on fresh medium. We use the term antagonistic to indicate that phenazine production counteracts the killing efficiencies of antibiotics applied to biofilms exogenously. We note that this definition of antagonism is not in line with classic definitions from the clinical drug–drug interaction field[22], which rely on conditions not directly applicable to our biofilm system (e.g. MIC testing in liquid culture, where the protective effect of phenazines is diminished). The effect of phenazines on ciprofloxacin tolerance is biofilm-specific, as it was low or undetectable in liquid-culture experiments.

**Protection from antibiotics is linked to metabolism.** Phenazines have various effects on *P. aeruginosa* biofilm physiology, some of which could affect survival during exposure to antibiotics (Fig. 2a): (1) they inhibit production of matrix, the exopolysaccharide scaffold that can support biofilm structure formation[23]; (2) they induce expression of efflux pumps[21]; and (3) they affect flux through central metabolism and balance the intracellular redox state[9,10]. To assess whether matrix or efflux pump production contribute to the antagonistic effect of phenazines on ciprofloxacin, we measured the in-biofilm survival of mutant strains after antibiotic exposure. To test the contribution of matrix production, we used a strain background lacking genes for production of the exopolysaccharide Pel, which is the main polysaccharide component of the biofilm matrix in PA14 (Δpel)[24,25]. To test the contribution of efflux pumps, we used a strain lacking the operons *mexGHI-opmD*, *mexPQ-opmE*, and

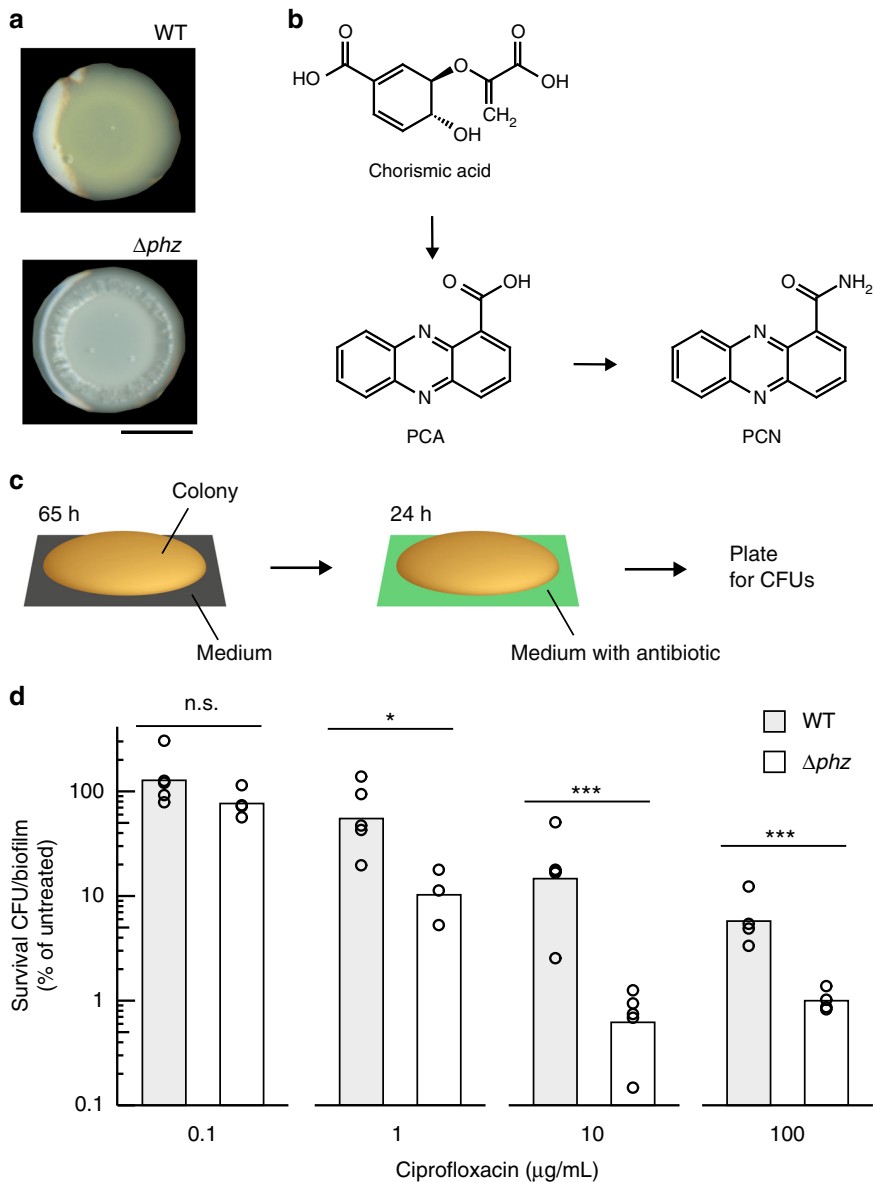

**Fig. 1** Cells from phenazine-null biofilms show increased sensitivity to ciprofloxacin. **a** Four-day-old colony biofilms of PA14 WT and the phenazine-null mutant (Δ*phz*) grown on a defined medium containing 20 mM glucose. Scale bar is 5 mm. **b** Schematic representing the biosynthesis of phenazines produced by glucose-grown PA14 biofilms. PCA phenazine-1-carboxylic acid, PCN phenazine-1-carboxamide. **c** Schematic of experimental design used to quantify antibiotic tolerance in colony biofilms. **d** Survival of WT and Δ*phz* cells in biofilms exposed to ciprofloxacin at four concentrations. Each count is normalized to the CFU count reached without antibiotics (which corresponds to 100%). Data for growth without antibiotics does not show significant differences between strains (Supplementary Figure 3). Each data point is a biological replicate, bar height indicates the mean of these replicates. *p* values are based on two-sided unpaired *t*-tests (n.s., not significant; *$p \leq 0.05$; ***$p \leq 0.001$)

*mexVW* (Δ*mex*). *mexGHI-opmD* and *mexPQ-opmE* encode RND efflux pumps that are upregulated by phenazines[21], while *mexVW* encodes the closest homolog to *mexHI*[26]. We found that phenazine-mediated ciprofloxacin tolerance was maintained in both of these strain backgrounds (Fig. 2b), indicating a limited effect of expression of efflux pumps or matrix production. However, when biofilms were grown on succinate, a carbon source that enters central metabolism downstream of glucose, the protective effect of phenazines was abolished (Fig. 2b). Influencing metabolism by altering the carbon source therefore has a stronger effect on phenazine-mediated tolerance than matrix production or efflux, suggesting that ciprofloxacin antagonism is linked to the effect of phenazines on metabolism and redox-balancing.

**SRS imaging reveals phenazine-dependent metabolism in biofilms.** The relationship between metabolic status and susceptibility to antibiotics has emerged as a prominent theme in recent literature, especially in the contexts of redox balancing and/or effects on the proton motive force[27–31]. For phenazines, the metabolic effects are potentially most significant in biofilm subzones where cells are limited by the electron acceptor oxygen and rely on phenazine reduction as an alternate redox-balancing strategy[10,20]. We thus used microsensors and microelectrodes to measure oxygen and extracellular redox potential, respectively, across depth in colony biofilms. As observed previously for biofilms grown on tryptone, we found that oxygen was depleted and became undetectable at a depth of ~70 μm (Fig. 3a)[10,20]. We also found that cells across depth in these biofilms carry out reduction

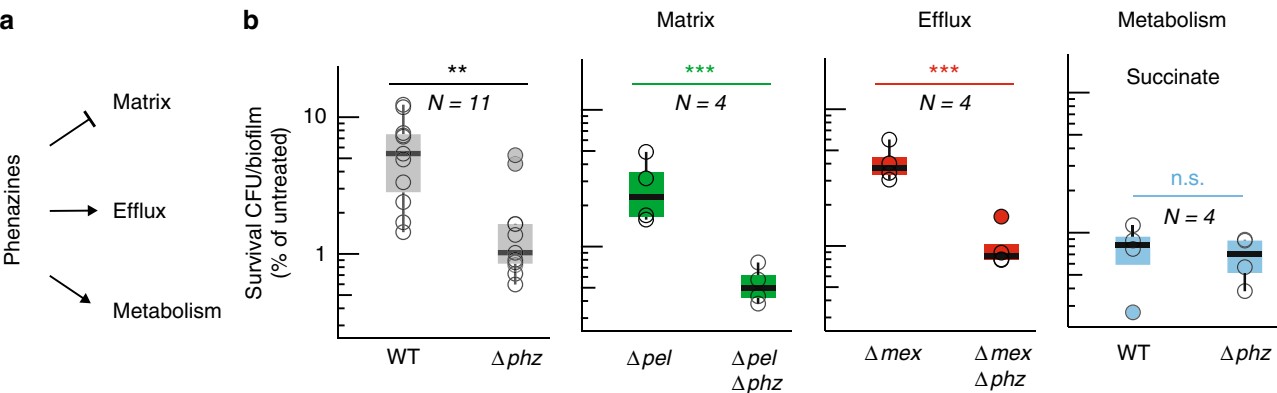

**Fig. 2** Phenazine-mediated protection does not require matrix or major efflux pumps but depends on carbon source. **a** Overview of known effects of phenazines on bacterial physiology. Phenazines can inhibit matrix production and induce expression of efflux pumps. Also, phenazines can alter flux through central metabolism and oxidize the cellular redox state. **b** Quantification of ciprofloxacin tolerance observed for cells from biofilms representing various genetic or environmental conditions. The data for the parent strains (WT and Δphz) grown with glucose as the carbon source are shown in gray in the left panel. Colored plots show the indicated mutant strains (green, red) or growth on succinate (blue). Each data point (N) is a biological replicate. $p$ values are based on unpaired one-sided $t$-tests (n.s., not significant; **$p \leq 0.01$; ***$p \leq 0.001$). Δpel corresponds to ΔpelB-G, Δmex to the triple mutant ΔmexGHI-opmD ΔmexVW ΔmexPQ-opmE (see table S1). The center line of the boxplot shows the median, the lower and upper hinges correspond to the first and third quartiles, and the whiskers extend to the most extreme points, limited to 1.5 times the interquartile range

of phenazines[20] (Fig. 3a, compare WT to Δphz). Interestingly, in contrast to results obtained for tryptone-grown biofilms[20], we observed that growth on glucose supported reduction of phenazines across the whole biofilm, including the oxygen-rich region.

The links we observed between metabolic status and the antagonistic effect of phenazines suggested to us that metabolic heterogeneity, induced by resource gradients, between biofilm subpopulations may lead to differential susceptibility to antibiotic treatment. To characterize metabolic heterogeneity, previous biofilm studies have mostly relied on the expression of inducible or unstable fluorescent proteins as readouts for metabolic activity as a function of depth[32–34]. Here, we employed a technique that quantifies biosynthetic activity more globally as incorporation of stable isotopes into biomass (such as proteins, lipids, and carbohydrates). Stable isotopes like deuterium have previously been used to study metabolism by mass spectrometry and Raman spectroscopy in an unbiased and minimally perturbing way in single bacterial and eukaryotic cells[35–37]. To directly examine metabolism in colony biofilms in a spatially resolved manner, we coupled stable isotope labeling with the emerging SRS microscopy[38,39]. SRS microscopy is a nonlinear optical imaging technique that provides a $10^8$-fold enhancement in spontaneous Raman scattering signal and $10^3$–$10^6$ times higher imaging speed than conventional Raman microscopy[40,41]. Using deuterium labeling, we can directly visualize the global metabolic activity as incorporated deuterium signal in colony biofilms with high sensitivity and specificity through SRS imaging of carbon–deuterium bonds (C–D) in the cell Raman-silent window (Supplementary Figure 6). A deuterium incorporation level as low as 0.1% of total biomass can be detected[42,43]. Compared to other Raman imaging techniques, SRS has a well-preserved spectrum that is free from non-resonant background, has linear dependence on the analyte concentration for quantitative analysis and endows 3D-sectioning capability[41].

We confirmed the robustness of our method by separately examining deuterium incorporation from two different substrates, D7-glucose and $D_2O$, via SRS imaging of live biofilms by optical sectioning (Fig. 3b and Supplementary Figure 7). We also compared optical sectioning to an alternate method in which colonies are subjected to paraffin-embedding and thin sectioning[44] ("paraffin sectioning", Fig. 3b) and SRS imaging is performed on the 10-μm-thin sections. All techniques yielded

qualitatively similar results, and although we observed considerable variation in the absolute deuterium signal between experiments, the relative distribution patterns were reproducible (Supplementary Figure 8a). Both WT and Δphz colonies showed peaks of metabolic activity at a depth of ~30–40 μm, while activity in the bottom third of biofilms (>100 μm depth) was below detection in both strains (Fig. 3b and Supplementary Figures 7, 8), consistent with previous studies describing a dichotomy between metabolically active cells at the biofilm–air interface and inactive cells at the biofilm–substrate interface[32–34,45]. Our results are unique, however, in that they show a complex distribution of metabolically active cells that is influenced by the presence of phenazines. Most notably, the metabolic patterns for WT and Δphz biofilms differed in that WT samples displayed a more prominent peak of activity below 50 μm. Our microsensor measurements indicate that the second population visible in the WT is located in an oxygen-depleted region of the biofilm where extracellular phenazines are in the reduced state (Fig. 3a). Finally, we also used SRS imaging and fluorescence microscopy to analyze biofilms containing a GFP-based reporter for the intracellular presence of oxidized phenazines[46], which showed maximal expression in the metabolically active, hypoxic region (Supplementary Figure 9d). The co-localization of reduced (Fig. 3a) and oxidized phenazines indicates that cells are catalyzing their redox cycling. Phenazines have been shown to accept metabolic electrons and facilitate redox balancing, ATP production and survival in $P.\ aeruginosa$[9,18,47]; these physiological effects could contribute to the maintenance of phenazine-dependent metabolic activity we observe via SRS imaging of deuterium incorporation.

To further investigate the relationship between phenazine-dependent metabolism and ciprofloxacin efficacy, we exposed biofilms to ciprofloxacin in the presence of $D_2O$ labeling (Fig. 3c and Supplementary Figure 8b). The protocol used for deuterium labeling was modified to mirror the setup used for measuring ciprofloxacin tolerance (Fig. 1c), i.e., colonies were transferred to the $D_2O$-containing medium 12 h earlier than for the experiments shown in Fig. 3b and incubated for 24 h after the transfer. In contrast to Fig. 3b, the modified protocol yielded similar metabolic activity profiles for WT and Δphz in the absence of ciprofloxacin (Fig. 3c). We attribute this discrepancy to the longer incubation time with $D_2O$, which allows both strains to reach

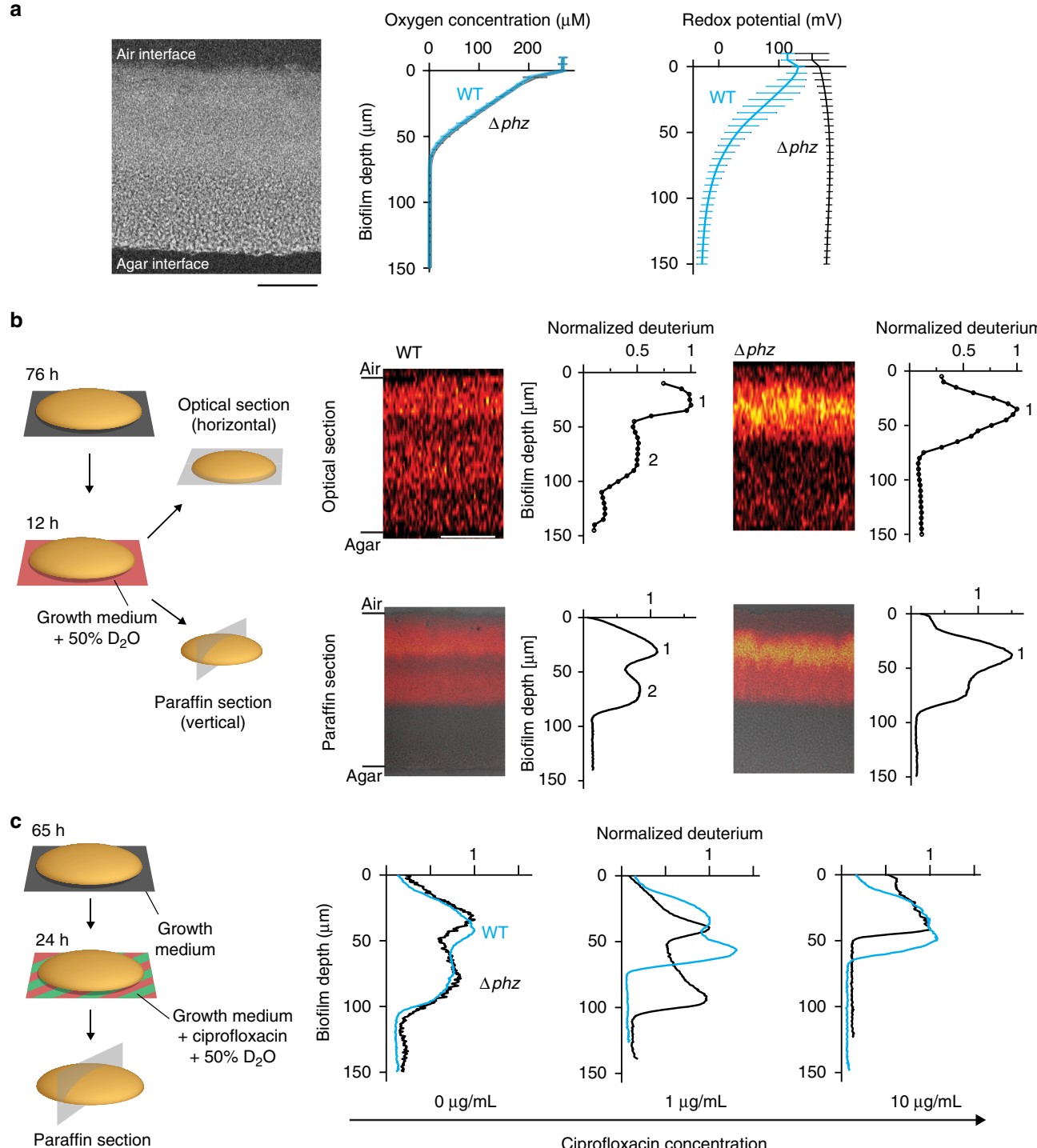

steady-state deuterium incorporation and diminishes differences in metabolic turnover rates. Nevertheless, incubation on 1 μg/ml ciprofloxacin yielded different labeling patterns for WT and Δ*phz* colonies (Fig. 3c). We were particularly intrigued by the emergence of distinct activity peaks in hypoxic biofilm regions for both WT and Δ*phz* colonies. While metabolic activity in the presence of phenazines is maximized at ~60 μm (in WT), in their absence activity peaks at ~90 μm (in Δ*phz*). The activity in Δ*phz* might arise from the enhancement of phenazine-independent redox-balancing mechanisms, such as high-affinity terminal oxidases that function at low oxygen concentrations, and/or

fermentation[9] (see also Fig. 4). We also note that the activity at 90-μm depth in Δ*phz* is susceptible to 10 μg/ml ciprofloxacin. Although we cannot definitively identify the biofilm subpopulations that are responsible for the differential tolerances of ciprofloxacin observed for WT and Δ*phz* biofilms (Fig. 1d), our findings provide insight into how antibiotics influence metabolic activity in situ, and show that this relationship is affected by phenazines. Also, these data highlight that the largest changes in metabolic profiles induced by ciprofloxacin and phenazines are visible at depth, i.e. their effects are strongest in oxygen-limited biofilm regions.

  

**Fig. 3** The distribution of metabolic activity in biofilms is influenced by phenazine production and exposure to ciprofloxacin. **a** Left: Section of a WT colony biofilm, prepared by paraffin embedding, stained with DAPI, and visualized by fluorescence microscopy. Scale bar is 50 μm. Data from sectioning experiments were collected at the approximate center of the colony in an area of 254 × 254 μm. Center and right: Microsensor and microelectrode profiling show that oxygen concentration is depleted at ~70 μm in WT (blue) and Δphz (black) biofilms (center) and that phenazines are reduced at depth in WT biofilms (right). Data show mean and standard deviation for biological replicates for oxygen (WT: N = 7, Δphz: N = 8) and for redox (WT: N = 8, Δphz: N = 3) microprofiling. **b** Left: Schematic of experimental design used to visualize metabolic activity in colony biofilms by stable isotope labeling. Spatially resolved readouts were acquired by either collecting images in 5 μm steps in z-direction in an area of 254 × 254 μm over the biofilm depth (optical sectioning) or by subjecting biofilms to paraffin embedding and sectioning, followed by imaging signal in a 10-μm-thin slice of the colony (paraffin sectioning). Right: Images and plotted deuterium signals obtained for optical and paraffin sections of colony biofilms after a 12-h incubation on D$_2$O-containing medium. Data plots show mean deuterium signal per biofilm depth. One replicate each of WT and Δphz is shown and is representative of at least five biological replicates. For data of all replicates, see Supplementary Figure 8a. Deuterium signals are normalized to the signal in peak 1 within each sample. Scale bar is 50 μm. Paraffin section images are overlaid with protein signal to outline the colony. **c** Left: Schematic of experimental design used to visualize metabolic activity after incubation on labeled medium containing ciprofloxacin. Right: Deuterium signal for one biological replicate each of WT (blue) and Δphz (black) after a 24 h incubation on medium containing D$_2$O and 0, 1, or 10 μg/ml ciprofloxacin. Deuterium signals are normalized to the signal in peak 1 within each sample. Data plots show mean deuterium signal per biofilm depth. For data of all replicates (N = 3), see Supplementary Figure 8b

**Cco complexes support phenazine-mediated antibiotic resistance.** Our observations suggest that phenazines support metabolic activity in oxygen-depleted biofilm subzones and that metabolic state influences the antibiotic susceptibility of cells in biofilms. To identify pathways that could enhance the survival of biofilm cells during antibiotic exposure, we tested mutants representing unique branches of energy metabolism: Δcco1cco2 and ΔldhA. Δcco1cco2 lacks the major terminal oxidases that catalyze O$_2$ reduction (i.e., respiration) and that are required for phenazine reduction in *P. aeruginosa* biofilms[20], while ΔldhA lacks an enzyme that converts pyruvate to lactate during survival by fermentation (Fig. 4a). Measurement of survival for cells from Δcco1cco2 biofilms revealed that the cbb$_3$-type terminal oxidases (i.e., Cco complexes) contribute to ciprofloxacin tolerance when phenazines are produced but not in the phenazine-null background (Fig. 4b). Genetic complementation confirmed that altered survival of Δcco1cco2 biofilms can be attributed to the function of this locus (Supplementary Figure 10a). Furthermore, the difference in survival between WT and Δcco1cco2 cannot be attributed to effects on phenazine production, because phenazine measurements for Δcco1cco2 biofilms yielded results that were similar to those for the WT (Supplementary Figure 10b). When we applied our SRS imaging technique to Δcco1cco2 biofilms, we found that, in agreement with a phenazine-dependent role for Cco terminal oxidases, the Δcco1cco2 mutation led to complete loss of the lower peak of metabolic activity (50–90 μm biofilm depth) that is visible in WT biofilms (Supplementary Figure 11). Also, we detected expression of both terminal oxidases at the corresponding depth (Supplementary Figure 9a, b). The peak of activity in the oxygen-depleted zone could thus be attributed to Cco-dependent phenazine reduction, indicating that this type of metabolism contributes to ciprofloxacin tolerance. These observations suggest that the previously described role of the cbb$_3$-type terminal oxidases in reducing phenazines[20] supports a metabolic state that contributes to ciprofloxacin tolerance in biofilms.

We next tested the contribution of pyruvate fermentation to antibiotic resistance by measuring survival upon ciprofloxacin treatment for cells from ΔldhA biofilms. ΔldhA biofilms showed a modest decrease in resistance that was not statistically significant (Supplementary Figure 12). However, as previous studies from our group have indicated that colonies grown on a complex medium containing tryptone and pyruvate carry out pyruvate fermentation[15], we sought to test whether this metabolism is operating in the biofilms grown on the defined, glucose-containing medium used here. We grew colonies of reporter strains that express *gfp* under the control of a promoter that is induced by lactate and examined thin sections by fluorescence

microscopy. We observed GFP fluorescence throughout both biofilms but saw maximal levels in their microaerobic zone, particularly in the Δphz background (Fig. 4c). These results indicate that electron acceptor-limited cells in biofilms route a portion of the glucose provided in the medium to lactate, possibly as a redox-balancing mechanism, and are consistent with previous observations of stationary-phase liquid cultures grown with glucose as the sole carbon source[9]. The more pronounced role of this metabolism in the Δphz background could account for the relatively subtle effect of the ΔldhA mutation that we observed in the WT (i.e. phenazine-producing) background. More broadly, it supports a model in which respiratory, rather than fermentative, metabolism is primarily responsible for the phenazine-dependent ciprofloxacin tolerance observed for cells in PA14 biofilms (Fig. 4c).

## Discussion

Previous literature describing metabolic heterogeneity in biofilms has generally differentiated between a metabolically active region at the oxygen-exposed interface and an inactive region at depth, where oxygen is limiting[32–34,45]. Based on our data from *P. aeruginosa* PA14 biofilms, the hypoxic region itself is metabolically diverse. In this zone (below 60-μm depth, where oxygen becomes undetectable), cells reduce pyruvate (Fig. 4c) and express high affinity terminal oxidases in parallel[20] (Supplementary Figure 9a, b). The presence of phenazines further expands the metabolic versatility in this region and leads to the formation of a distinct metabolically active subpopulation that we detected by stable isotope labeling and SRS imaging (Fig. 3b).

Furthermore, our data suggest that metabolic versatility in redox balancing contributes to tolerance to ciprofloxacin. We propose that Cco-mediated phenazine reduction constitutes a redox-balancing pathway that confers a physiological condition of enhanced ciprofloxacin tolerance (Fig. 4d), in line with previous reports highlighting links between respiration and antibiotic tolerance[27,29,31]. We note that, while the Δcco1cco2 mutant shows decreased antibiotic tolerance relative to the WT, it nevertheless shows survival levels that are higher than that of the Δphz mutant, indicating that additional mechanisms contribute to the antagonistic effect of phenazines (Fig. 4b). In the presence of phenazines, pyruvate fermentation is attenuated (Fig. 4c), highlighting the role of phenazines in determining the metabolic organization of different subpopulations within a biofilm (Fig. 4d). For aminoglycoside antibiotics like tobramycin, for which we also observed an antagonistic effect of phenazines, reduction of the proton motive force as a result of respiration has been shown to protect cells by diminishing drug uptake[27], though

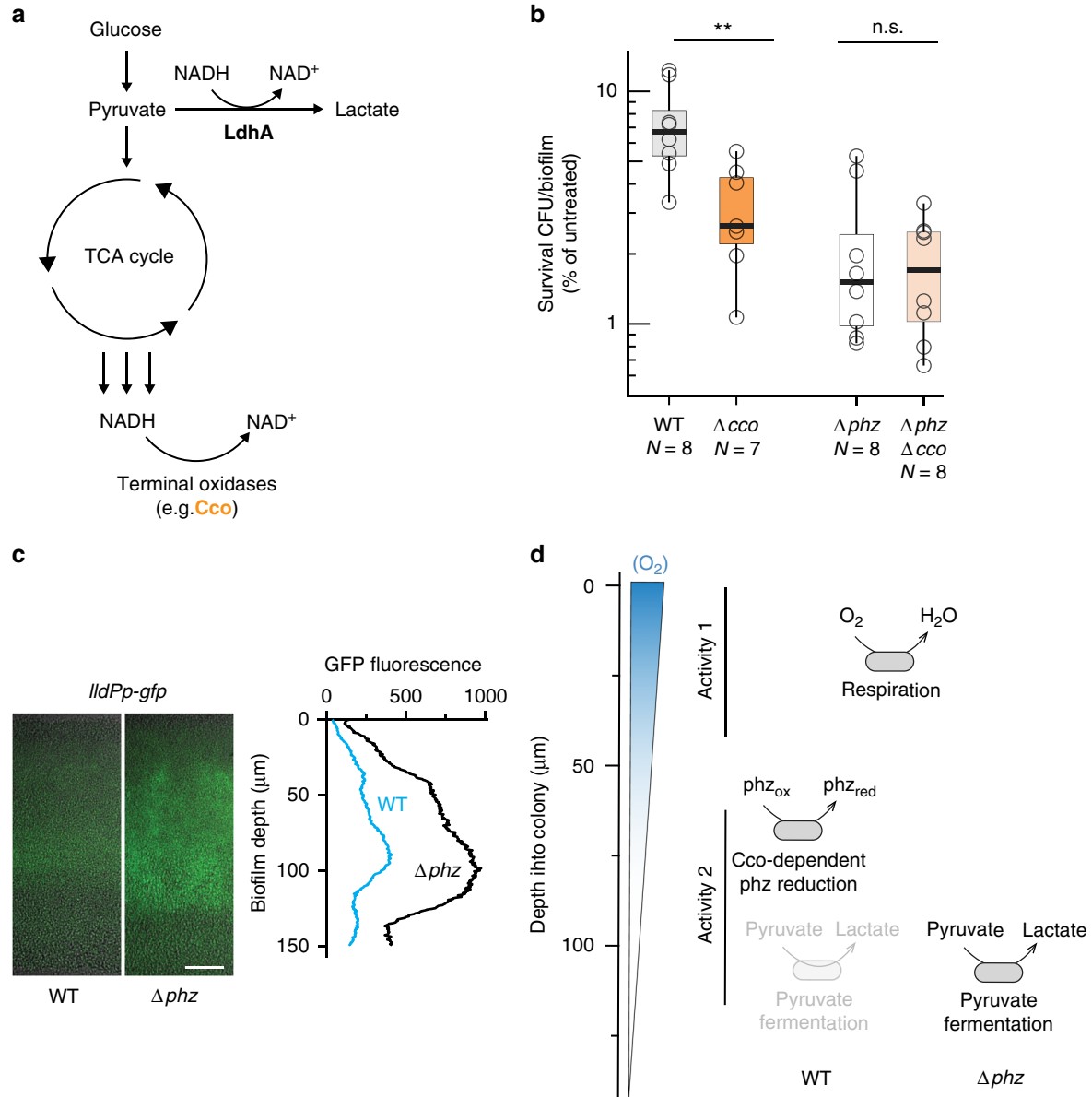

**Fig. 4** Analyses of antibiotic sensitivity and gene expression indicate that diverse redox-balancing pathways are functioning in PA14 biofilms. **a** Overview of the redox-balancing pathways investigated. NADH can be re-oxidized by pyruvate fermentation via LdhA or by the electron transport chain via terminal oxidases, such as the $cbb_3$-type Cco complexes implicated in phenazine reduction. **b** Ciprofloxacin (100 μg/mL) tolerance observed for cells from biofilms formed by *cco* mutants. Data for the parent strains (WT and Δ*phz*) are shown in gray. *p* values are based on an unpaired two-sided *t*-test between strain pairs as indicated (n.s., not significant; **$p \leq 0.01$). Data for growth without antibiotics does not show significant differences between strains (Supplementary Figure 3). The center line of the boxplot shows the median, the lower and upper hinges correspond to the first and third quartiles, and the whiskers extend to the most extreme points, limited to 1.5 times the interquartile range. **c** Expression analyses of WT and Δ*phz* colony paraffin sections show lactate production (which activates expression of the *lldPDE* operon) in biofilms grown on defined medium with glucose as the sole carbon source. One representative biological replicate is shown (data for all replicates (*N* = 3) is shown in Supplementary Figure 9e). Scale bar is 25 μm. **d** Model depicting the metabolisms that could support redox balancing in oxic (activity 1) and hypoxic (activity 2) biofilm subzones, contributing to activities detected by isotope labeling/SRS imaging and to antibiotic tolerance

currently the mechanistic basis whereby respiration supports ciprofloxacin tolerance is less clear.

Our results represent the first direct visualization of the heterogeneous distribution of metabolism inside biofilms by in situ SRS metabolic imaging of stable isotope incorporation. This technique can be generally applied for studying microbial metabolism and antibiotic treatment in complex settings with high spatial resolution and minimal perturbation, which is of great importance considering that biofilms are one of the main contributors to persistent and antibiotic-resistant infections[48]. In addition, our data suggest that treatment of *P. aeruginosa* biofilm infections is influenced by interactions of antibiotics and phenazines, compounds detectable in cystic fibrosis patients[49]. Our findings thus highlight the interactions between small molecule metabolites, primary metabolism, and antibiotics that can impact the survival of microbes that cause biofilm-based infections.

## Methods

**Bacterial strains and growth conditions**. Strains and plasmids used are listed in Supplementary Tables 1 and 2. Biological replicates were started from a single

colony streaked out from a frozen glycerol stock on lysogeny broth agar (LB;[50] 1% tryptone, 1% NaCl, 0.5% yeast extract, 1% agar). Colonies were inoculated in 2 ml LB and grown for 12–13 h (37 °C, shaking at 250 rpm). Cells were subcultured for around 4 h in 20 mM glucose MOPS minimal medium (50 mM 4-morpholinepropanesulfonic acid (pH 7.2), 43 mM NaCl, 93 mM NH$_4$Cl, 2.2 mM KH$_2$PO$_4$, 1 µg/ml FeSO$_4$·7H$_2$O, 1 mM MgSO$_4$·7H$_2$O) in a 1:100 dilution until they reached exponential phase, with an optical density at 500 nm between 0.25 and 0.5. The terminal oxidase mutants (Δcco1cco2) grew slower in subculture and were hence started with a dilution of 1:60. Optical density was adjusted to 0.25 with phosphate buffered saline (PBS), and cells were washed (centrifugation for 5 min, 6800 × g) and resuspended in PBS for further use. For most experiments, 5 µl cells were consequently spotted on 1% agar-solidified media and incubated at 25 °C and > 90% humidity to form colony biofilms.

**Strain construction.** Strains containing markerless deletions in efflux pumps and Cco1 and Cco2 terminal oxidases were made by homologous recombination[51]. In brief, 1 kb flanking sequence was amplified from each side of the target gene (for primers, see Supplementary Table 3) and inserted into the plasmid pMQ30 using the yeast gap repair method in Saccharomyces cerevisiae InvSc1[52]. The plasmid was moved into PA14 by biparental mating with E. coli strain UQ950. Following initial selection on 100 µg/ml Gentamicin, markerless mutants resulting from double recombination were further selected on 10% sucrose LB plates without NaCl. Mutants were confirmed by PCR. Multiple deletions were generated stepwise by using strains already containing mutations as host for biparental mating. The cco1cco2 complementation strain was made in a similar manner: the coding sequences of cco1 and cco2 were cloned, verified by sequencing, and inserted at the deletion site.

**Use of high-pressure liquid chromatography to quantify phenazines.** To extract phenazines from biofilms, colonies were grown on filter paper overlaid by a thin 1% agar layer as for the CFU count experiments. Phenazines were extracted from this filter paper, as well as the agar below the colony (with a volume of 6 ml) by nutating the biofilm and the agar in 5 ml HPLC-grade methanol overnight at room temperature in the dark. Phenazines from liquid culture were directly quantified in the supernatant, from which cells had been removed by centrifugation (5 min, 16,870 × g). Three hundred microliters of supernatant or phenazine extract were filtered through a 0.22 µm cellulose Spin-X column (Thermo Fisher Scientific 07-200-386) and 200 µl of the flow-through were loaded into an HPLC vial. Phenazines were quantified using reversed-phase high-performance liquid chromatography (Agilent [Santa Clara, CA] 1100 HPLC System) with a biphenyl column (Kinetex 00F-4622-E0, 4.6 × 150 mm, 2.6 µm). A gradient method was used with (a) deionized water (containing 0.02% formic acid) and (b) methanol (containing 0.02% formic acid) by increasing (b) from 40% to 100% within 25 min with a flow rate of 0.4 ml min⁻¹ at room temperature, followed by a hold at 100% methanol for 5 min. Absorption was quantified at 366 nm. The identity of phenazine peaks was verified by the absorption spectrum, as well as comparison with the retention time of phenazine standards.

**Quantification of ciprofloxacin tolerance in biofilms.** To start biofilm colonies, 5 µl washed cell culture, prepared as explained above, was spotted onto 20 mM glucose MOPS minimal medium with 1% agar (40 ml in a 100 mm × 15 mm Petri dish). Cells were spotted onto a filter disk (diameter: 25 mm; pore size: 0.2 µm; GE Healthcare 110606) that was covered with a thin (~1 mm high) 1% agar layer to reduce effects of the filter on colony morphology. For survival tests on succinate, 20 mM sodium succinate hexahydrate was used. Biofilms were incubated at 25 °C with >90% humidity (Percival [Perry, IA] CU-22L). Colony images were obtained with a flatbed scanner (Epson [Japan] E11000XL-GA). Colonies were incubated for around 3 days (64–65 h) and then moved with the filter to a 35 × 10 mm Petri dish (VWR 25373-041) containing 6 ml of 20 mM glucose MOPS minimal medium and 1% agar as well as antibiotics. Ciprofloxacin (Sigma-Aldrich 17850) was dissolved in acidified sterile water and stocks were stored at −20 °C. Carbenicillin disodium salt (Teknova, C2105) was dissolved in sterile water and stocks were stored at −20 °C. Tobramycin sulfate (VWR AAJ62995-03) and colistin sulfate (VWR 10791-860) were dissolved in sterile water and directly used. Biofilms were exposed to ciprofloxacin for 24 h at 25 °C with >90% humidity and then homogenized in 1 ml PBS using a bead mill homogenizer (Omni [Kennesaw, GA] Bead Ruptor 12; at high setting for 99 s) and ceramic beads (Thermo Fisher 15 340 159, diameter of 1.4 mm). The cell suspension was serially diluted in PBS, plated onto 1% tryptone plates and incubated incubated at 37 °C for 24 h, then at 25 °C for 24 h before CFU counting.

**Quantification of ciprofloxacin tolerance in stationary phase.** To quantify survival of cells to ciprofloxacin in stationary phase liquid culture, cells were grown in 50 ml 20 mM glucose MOPS minimal medium in a 250-ml flask for 16 h, shaking at 250 rpm at 37 °C. Cultures were started with a 1:50 dilution from the washed subculture prepared as described above. After 16 h, when cells had reached stationary phase, ciprofloxacin was added to the cultures, and samples were taken for CFU counts at 0 and 4 h to quantify survival over time. Cells were serially diluted in PBS and plated onto 1% tryptone plates for CFU counting. To quantify

the effect of phenazines on ciprofloxacin-exposed liquid culture, cells were grown in a 96-well plate as explained above. Pyocyanin standard (dissolved in DSMO; Cayman Chemical 10009594, >98%) and PCA standard (dissolved in DMSO; Apexmol, 95%) were stored at −80 °C and added to the wells prior to inoculation.

**Determination of minimal inhibitory concentration.** To determine the minimal inhibitory concentration (MIC) cells were grown in a clear, flat-bottom polystyrene 96-well plate (Greiner Bio-One 655001). Starting concentrations for testing the MIC were based on literature values[45,53–55]. MIC was determined as the lowest antibiotic concentration tested at which blanked optical density at 500 nm stayed below 0.1 and no clumps had formed after an incubation for 24 h in 20 mM glucose MOPS minimal medium at 37 °C. Growth was quantified with a plate reader (Biotek Synergy H1, linear continuous shaking with a frequency of 731 cycles per minute). Cultures were started from a 1:100 dilution of the washed subculture, prepared as explained above. Growth was quantified as optical density at 500 nm, read every 10 min for 24 h.

**Spontaneous Raman spectroscopy.** Raman spectra of biofilm thin sections were collected on a confocal Raman microscope (Xplora, Horiba) using the LabSpec 6 software. The samples were excited by a 532 nm diode laser through a 50X air objective (Mplan N, 0.75 NA, Olympus) at room temperature. The power was 27 mW after the objective and the acquisition time for the spectra was 20 s.

**SRS microscopy.** An integrated laser source (picoEMERALD, Applied Physics & Electronics, Inc.) was used to produce both a Stokes beam (1064 nm, 6 ps, intensity modulated at 8 MHz) and a tunable pump beam (720–990 nm, 5–6 ps) at a 80 MHz repetition rate. The spectral resolution of SRS is FWHM = 6–7 cm⁻¹. Two spatially and temporally overlapped beams with optimized near-IR throughput were coupled into an inverted multiphoton laser-scanning microscope (FV1200MPE, Olympus). Both beams were focused on the cell samples through a 25X water objective (XLPlan N, 1.05 N.A. MP, Olympus) and collected with a high N.A. oil condenser lens (1.4 N.A., Olympus) after the sample. By removing the Stokes beam with a high O.D. bandpass filter (890/220 CARS, Chroma Technology), the pump beam is detected with a large area Si photodiode (FDS1010, Thorlabs) reverse-biased by 64 DC voltage. The output current of the photodiode was electronically filtered (KR 2724, KR electronics), terminated with 50 Ω, and demodulated with a RF lock-in amplifier (SR844, Stanford Research Systems) to achieve near shot-noise-limited sensitivity. The stimulated Raman loss signal at each pixel was sent to the analog interface box (FV10-ANALOG, Olympus) of the microscope to generate the image. All images were acquired with 30 µs time constant at the lock-in amplifier and 100 µs pixel dwell time (~7 s per frame of 256 × 256 pixels). Measured after the objectives, 12 mW pump power and 40 mW Stokes power were used to image the protein CH$_3$ 2940 and off-resonance 2650 cm⁻¹ channel. 24 mW pump beam and 120 mW Stokes beam were used to image the carbon–deuterium 2165, 2175, and off-resonance 2000 cm⁻¹ channels.

**Stable isotope labeling.** To image metabolic activity as incorporation of deuterium isotopes, 20 mM glucose MOPS minimal medium was amended with either 20 mM deuterated D7-glucose (Sigma Aldrich) or 50% deuterated water (Sigma Aldrich; 2 ml volume in a 35 × 10 mm Petri dish). For pulse experiments with deuterated water, biofilms were grown on unlabeled medium for 76 h or, in the case of the antibiotic tests for 64 h, followed by incubation on medium with 50% D2O for 12 or 24 h, respectively. For optical sectioning, deuterated water in the media was removed by incubation on 1% agar with unlabeled H$_2$O for 30 min prior to SRS imaging. Chase experiments were conducted by growing biofilms for 3 days on MOPS minimal medium containing 20 mM deuterated D7-glucose, which was then chased by incubation on 20 mM glucose MOPS minimal medium for 12 h.

**Preparation of biofilms for SRS imaging via optical sectioning.** Colonies were grown on a 1.5% thin agar layer on top of a filter in media described above. After deuterium labeling, the colony was transferred onto a coverslip using the thin agar layer. Spacers (Sigma Aldrich) were used to create an imaging chamber with a microscopy glass slide on top of the spacer for SRS imaging of live biofilms.

**Paraffin-embedded thin sectioning for imaging.** Thin sectioning was performed similar to as previously described[44]. Colony biofilms were moved onto a two-layer agar plate using the thin (~1 mm high) 1.5% agar layer biofilms had grown on. The two-layer agar consisted of a bottom layer of 32 ml and a top layer of 8 ml of 1% agar in a 100 mm × 15 mm Petri dish. After transfer of the colony, the plate was covered with 8 ml of 1% agar. After polymerization of the agar, the embedded colony was cut out including the surrounding agar and pre-fixated at 4 °C in 50 mM L-lysine hydrochloride and PBS, followed by fixation in 50 mM L-lysine hydrochloride, PBS, and 4% paraformaldehyde, first for 4 h at 4 °C and then at 37 °C for 24 h in the dark. Dehydration, sectioning to 10 µm-thin sections, and rehydration were performed as described previously[44]. Sections were mounted in Tris-buffered DAPI:Fluorogel (Thermo Fisher Scientific 50-246-93) or Tris-buffered Fluorogel without DAPI (Thermo Fisher Scientific 50-247-04) for correlative SRS and fluorescence imaging. Fluorescence imaging was performed using

the Olympus FV1200 confocal microscope with standard laser excitation and bandpass filter set for each fluorescent reporter.

**Oxygen and redox gradient measurements**. Four-days old colony biofilms directly grown on 20 mM glucose MOPS minimal medium without filter were used for oxygen profiling and redox profiling as described previously[20].

**Image analysis**. For optical sections, images collected were imported into Fiji and deuterium and protein images from the same biofilm depth were manually aligned. Mean signal per height was exported as a csv file and further analyzed in R[56]. Data plots shown are based on protein-corrected deuterium signal.

For paraffin sections, signal profiles over height were assembled using a combination of Fiji[57] and R[56]. Raw images were imported in Fiji and rotated such that the bottom of the biofilm was aligned to the bottom of the image. A mask of the biofilm section was created based on either fluorescence (for fluorescence images) or the protein channel (for protein and deuterium signal). In cases where no masks could be generated by thresholding, the mask was manually drawn around the biofilm section. Raw data from within this mask were exported as csv and further analyzed in R. In a custom-written R script, the biofilm section was aligned at the top interface of the biofilm and average signals per height were calculated. Images of paraffin sections show background subtracted-deuterium signal overlaid with background-subtracted protein channel (whereby we target the methyl group vibration in proteins with a frequency of 2940 cm$^{-1}$) to visualize the outline of the biofilm section. For fluorescence images, the background was subtracted as the average auto-fluorescence signal measured in a promoterless reporter control.

**Statistical analysis**. Statistical analyses were conducted with R[56]. Levene's test for homogeneity of variance was carried about for data subjected to $t$ tests.

## Data availability
The datasets generated during and/or analyzed during the current study are available from the corresponding authors on reasonable request.

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

## Acknowledgements

We thank Blanche Fields and Yu-Cheng Lin for help with generating mutant strains and William Cole Cornell for help with paraffin sectioning. This work was supported by an SNF Early Postdoc Mobility Fellowship (#PP2EZP3-162260) to K.T.S., a Raymond and Beverly Sackler Center Postdoc Fellowship to F.H., NIH training grant 5T32GM008798 to J.J., NSF GRFP grant DGE—1644869 to B.W., NIH/NIAID grant R01AI103369 and an NSF CAREER award to L.E.P.D. and grants from the NIH (R01 EB020892) and the Camille and Henry Dreyfus Foundation to W.M.

## Author contributions

K.T.S., F.H., W.M., and L.E.P.D. conceived and designed the study. K.T.S., S.N., and B.W. performed survival experiments. K.T.S. performed all paraffin sectioning and HPLC experiments, F.H. performed all imaging and conceived the deuterium labeling setup. J.J. performed the redox and oxygen gradient measurements. K.T.S., S.N., B.W., and F.H. analyzed the data. K.T.S., F.H., A.P.W. and L.E.P.D. wrote the paper with input from J.J., S.N., B.W., and W.M. All authors approved the final version.

## Additional information

**Competing interests:** The authors declare no competing interests.

