## [Peer Review File · Nature Communications]

Reviewers' comments:

Reviewer #1 (Remarks to the Author):

The manuscript, "Phenazine production promotes antibiotic tolerance and metabolic heterogeneity in *Pseudomonas aeruginosa* biofilms" describes a novel approach for assaying vertical metabolic heterogeneity in biofilms using stable isotope labeling and Raman scattering microscopy (SRS). Using this approach, the investigators labeled biofilms with deuterium, generated thin sections of the biofilms, then imaged the vertical sections using SRS. The investigators found that they could map the metabolic heterogeneity to a greater resolution than other approaches, and in particular, were able to map metabolism in hypoxic zones, where the cells used non-oxygen-based forms of respiration (phenazine utilization and pyruvate utilization). The zones of phenazine and pyruvate utilizations correlated well with oxygen profiles through the biofilms.

The investigators used this approach to characterize the role of phenazines (alternative electron acceptors produced by *P. aeruginosa*) on the tolerance of cells to antibiotics, focusing primarily on ciprofloxacin. The results showed that a phenazine mutant strain was more sensitive to most antibiotics than wild-type strain in biofilms but not in planktonic culture, suggesting that the phenazines antagonize antibiotic sensitivity. They tested the mode of action for the phenazines, by using various mutant strains, to determine if the role had to do with matrix production (PEL production) efflux (*mexA*), or metabolism (glucose versus succinate).

Overall, I find this manuscript to be thorough and well-written. It introduces a new approach to the study of biofilm metabolic activity, which will be useful for other investigators.

Minor comments:

Ln 71: The header states "Phenazine production protects biofilms from antibiotics." I feel like this is overstated. While there is an approximate 10-fold difference between the mutant and the wild-type, there is still 10% of the biofilm left – so it is not "protected".

Figure 2: This graph is complicated (visually). I don't know the purpose of the colored lines (since it is not a time-course study). Perhaps present the data in a more simplified manner.

Reviewer #2 (Remarks to the Author):

Schiessl and colleagues identify that phenazine, a natural product produced by *Pseudomonas aeruginosa*, antagonizes the effect of ciprofloxacin on biofilms. The authors demonstrate that phenazines do this by altering the metabolism in microaerophilic and anoxic regions of biofilms. The use of stable isotope probing and SRS imaging to look at metabolism in the colony biofilm model is especially elegant and will add to a growing body of work demonstrating heterogeneity in biofilm communities. While there is previous literature demonstrating links between metabolism (ex. active starvation responses, anoxic niches, etc.) and antibiotic tolerance, the additional link to redox-balancing via phenazine production is an intriguing, additional piece to the puzzle. It provides some insight into biofilm antimicrobial resistance mechanisms, which I believe will be of interest to a broad readership and biofilm microbiologists alike. The integration of genetic methods with the chemical imaging and susceptibility testing provides confidence in the results. The findings are exciting, and the manuscript is well-written.

Constructive criticism

- The term antagonism has a rigorous mathematical definition in the field of antibacterial

susceptibility testing. Synergistic or antagonistic interactions are established using the minimum fraction inhibitory concentration index (FICI) or minimum fraction bactericidal concentration index (FBICI) calculation. The authors should perform checkerboard assays and provide these data if the term antagonism is going to be used. Alternatively, a careful explanation of this term and how it is applied to the present work should be provided to satisfy those of us working in antibacterial research and development. Note that formula and standardized methods for these assays are widely available in the literature, and they do not require a great deal of work.

- While the small molecule-drug interactions provide confidence in the work, genetic complementation should still be used to analyze the phenotypes of *ldhA* and *cco* mutant phenotypes. The work is carefully executed in this manuscript; however secondary site mutations can unexpectedly occur in cell lines, and these secondary mutations can be responsible for surprising phenotypes. There are a variety of genetic methods available for complementation analysis of *P. aeruginosa* that are relatively facile.

Minor points:

- Line 211. The title provided for this section is not clear. Could the authors please rephrase it?

- Figure 1. Perhaps I have missed this, but are the cell numbers for wild type and Δ phz cells equal? In other words, the data in this figure are normalized and presented as relative survival vs. the control group with no antibiotic. However, are the control groups equal? This would eliminate any concern that inoculum effect might be a contributor to perceived changes in antibiotic sensitivity. Please note, however, that if the cell numbers are not equivalent in wild type and mutant biofilms, that the interpretation would likely be called into question by many in the field.

- Figure 2. Could the authors please provide the letter designations for the *pel* and *mex* genes that were deleted for this analysis? In the case of the engineered *mex* mutant bearing multiple mutations, this information might be better in the figure legend.

- Figure 4. Could the authors please maintain the same y-axis scale (i.e. range) in panels b and d to enable easier comparisons for the reader? Is it the same data in panel b as in panel d for the wild type group (it looks conspicuously similar)? Why not combine data in b and d into a single panel? Lastly, same criticism as for Figure 1 above – are there equivalent starting numbers of cells in wild type and mutant biofilms?

- Figure S4. It might be simpler to present this information in a line graph in a single figure panel. Was there a reason to split this information across four panels?

Joe Harrison, Associate Professor and Canada Research Chair, University of Calgary

Reviewer #3 (Remarks to the Author):

Review comments on

Ref.: NCOMMS-18-23186

Title: Phenazine production promotes antibiotic tolerance and metabolic heterogeneity in

Pseudomonas aeruginosa biofilms

by Prof Dietrich and co-worker

This manuscript described the application of microsensors and stimulated Raman scattering (SRS) microscopy to detect redox profiles and metabolic activity of cross-sectioned colonies. The author regarded colonies as biofilms, however, it is hard to equal colonies to biofilms. Actually it has been shown that colonies of *Pseudomonas aeruginosa* are more similar to planktonic cells instead of biofilm in terms of protein profiles (Mikkelsen et al., *J Bacteriol* (2007) 189:2411-2416). Although the work used a few techniques to measure the cross-section of the colonies, the findings are not robust and it is unclear why the cross-sectioned colonies are relevant to biofilm, given that colonies are not really biofilm. This manuscript may be more suitable to a specific journal.

Main concerns:

1. The deuterium from D₂O or deuterated D₇-glucose is able to integrate into cells within 20 minutes (Reference 35, Berry et al., PNAS 2015). It is not surprised that fresh cells on the surface of a colony lost deuterium after subsequently growing in 20mM unlabelled glucose for 12 hours (Fig. 5). What is the diffusion effect of D₂O and ciprofloxacin (Fig. 3)? What is the control result without D₂O?
2. One of important conclusion is that phenazines promote metabolic activity of cells in microaerobic biofilm. Without the distribution data of phenazines, it is hard to link phenazine production and ciprofloxacin effect across the section.
3. SRS is an important technique in the manuscript. Can the authors provide the information how SRS setting is used to detect C-D band. For example, the resolution of SRS spectra. What are original SRS spectra and statistical analysis?
4. Many data have no replicates and error bars (Fig. 3a, 3b and 3c, Fig. 4c).

Other comments

1. Fig. 3a, the finest tip of oxygen microsensor is 8-12 μm (<http://www.unisense.com/O2/>), how can the author produce oxygen profile with 5 μm resolution?
2. Figure 3b, how many replicates have been done and what is the errors? What is the controls?
3. What is the diffusion effect of phenazine when it is added externally. What is the diffusion effect of medium, glucose and D₂O?

We were pleased to learn that the reviewers found the manuscript to be interesting and well-written, and very much appreciate their constructive comments. We have responded to their suggestions by carrying out additional experiments and amending the text, and believe these changes have enhanced the manuscript.

Our point-by-point responses to reviewers' comments are provided below (in blue).

Thank you again for your time and consideration.

Sincerely,

Lars Dietrich

Reviewer #1 (Remarks to the Author):

The manuscript, "Phenazine production promotes antibiotic tolerance and metabolic heterogeneity in *Pseudomonas aeruginosa* biofilms" describes a novel approach for assaying vertical metabolic heterogeneity in biofilms using stable isotope labeling and Raman scattering microscopy (SRS). Using this approach, the investigators labeled biofilms with deuterium, generated thin sections of the biofilms, then imaged the vertical sections using SRS. The investigators found that they could map the metabolic heterogeneity to a greater resolution than other approaches, and in particular, were able to map metabolism in hypoxic zones, where the cells used non-oxygen-based forms of respiration (phenazine utilization and pyruvate utilization). The zones of phenazine and pyruvate utilizations correlated well with oxygen profiles through the biofilms.

The investigators used this approach to characterize the role of phenazines (alternative electron acceptors produced by *P. aeruginosa*) on the tolerance of cells to antibiotics, focusing primarily on ciprofloxacin. The results showed that a phenazine mutant strain was more sensitive to most antibiotics than wild-type strain in biofilms but not in planktonic culture, suggesting that the phenazines antagonize antibiotic sensitivity. They tested the mode of action for the phenazines, by using various mutant strains, to determine if the role had to do with matrix production (PEL production) efflux (*mexA*), or metabolism (glucose versus succinate).

Overall, I find this manuscript to be thorough and well-written. It introduces a new approach to the study of biofilm metabolic activity, which will be useful for other investigators.

We thank the reviewer for these positive comments.

Minor comments:

Ln 71: The header states "Phenazine production protects biofilms from antibiotics." I feel like this is overstated. While there is an approximate 10-fold difference between the mutant and the wild-type, there is still 10% of the biofilm left – so it is not "protected".

We agree and changed the header to “Phenazine production increases survival of cells in biofilms upon exposure to antibiotics” (lines 68-69).

Figure 2: This graph is complicated (visually). I don't know the purpose of the colored lines (since it is not a time-course study). Perhaps present the data in a more simplified manner.

We appreciate this feedback. We intended to highlight survival differences between WT and Δ phz with the differently sloped lines, but agree that this led to a cluttered appearance and was not self-explanatory. We have now changed the layout to the one used in Figure 4, omitting the lines and using boxplots instead. We hope that this will clarify the results.

Reviewer #2 (Remarks to the Author):

Schiessl and colleagues identify that phenazine, a natural product produced by *Pseudomonas aeruginosa*, antagonizes the effect of ciprofloxacin on biofilms. The authors demonstrate that phenazines do this by altering the metabolism in microaerophilic and anoxic regions of biofilms. The use of stable isotope probing and SRS imaging to look at metabolism in the colony biofilm model is especially elegant and will add to a growing body of work demonstrating heterogeneity in biofilm communities. While there is previous literature demonstrating links between metabolism (ex. active starvation responses, anoxic niches, etc.) and antibiotic tolerance, the additional link to redox-balancing via phenazine production is an intriguing, additional piece to the puzzle. It provides some insight into biofilm antimicrobial resistance mechanisms, which I believe will be of interest to a broad readership and biofilm microbiologists alike. The integration of genetic methods with the chemical imaging and susceptibility testing provides confidence in the results. The findings are exciting, and the manuscript is well-written.

We appreciate the reviewer's positive comments.

Constructive criticism

- The term antagonism has a rigorous mathematical definition in the field of antibacterial susceptibility testing. Synergistic or antagonistic interactions are established using the minimum fraction inhibitory concentration index (FICI) or minimum fraction bactericidal concentration index (FBCI) calculation. The authors should perform checkerboard assays and provide these data if the term antagonism is going to be used. Alternatively, a careful explanation of this term and how it is applied to the present work should be provided to satisfy those of us working in antibacterial research and development. Note that formula and standardized methods for these assays are widely available in the literature, and they do not require a great deal of work.

*We thank the reviewer for this valuable feedback and for bringing to our attention the rigorous, field-specific definition of the term “antagonism”. After careful consideration, we've concluded that for *P. aeruginosa* the phenazine-ciprofloxacin interaction does not fit with this definition. Although it is well-established that phenazines are toxic for diverse organisms,*

they are not toxic for P. aeruginosa (at least within the concentration range relevant for this study). This caveat also precludes us from performing the checkerboard assay. Therefore, we rephrased our description of the drug interaction in the updated manuscript and explain more carefully what we mean by “antagonistic interaction”, as follows in lines 100-105:

“We use the term antagonistic to indicate that phenazine production counteracts the killing efficiencies of antibiotics applied to biofilms exogenously. We note that this definition of antagonism is not in line with classic definitions from the clinical drug-drug interaction field (Odds 2003), which rely on conditions not directly applicable to our biofilm system (e.g. MIC testing in liquid culture, where the protective effect of phenazines is diminished).”

- While the small molecule-drug interactions provide confidence in the work, genetic complementation should still be used to analyze the phenotypes of *ldhA* and *cco* mutant phenotypes. The work is carefully executed in this manuscript; however secondary site mutations can unexpectedly occur in cell lines, and these secondary mutations can be responsible for surprising phenotypes. There are a variety of genetic methods available for complementation analysis of *P. aeruginosa* that are relatively facile.

*We appreciate the reviewer pointing this out and agree that genetic complementation is imperative to confirming the reported phenotypes for the Δcco and $\Delta ldhA$ mutants. We have now engineered a Δcco complementation strain and show that it phenocopies WT with respect to its sensitivity to ciprofloxacin, confirming that the *cbb₃*-type oxidases may contribute to ciprofloxacin tolerance in the presence of phenazines. However, we were unable to generate the *ldhA* complementation of the $\Delta phz\Delta ldhA$ strain due to technical difficulties. We have therefore removed our observations on the effects of fermentation-based redox activity on survival from the manuscript. Since we had initially found that this effect was phenazine independent, we feel that removing these data does not affect the main conclusion of the study, i.e. that phenazines support survival in the presence of ciprofloxacin and that the *Cco* terminal oxidases contribute to this effect.*

Minor points:

- Line 211. The title provided for this section is not clear. Could the authors please rephrase it?

We agree and changed the previous title, “Redox-balancing pathways promote survival of antibiotic treatment”, to “Cco terminal oxidases contribute to phenazine-dependent survival of ciprofloxacin-treated biofilms” (lines 224-225).

- Figure 1. Perhaps I have missed this, but are the cell numbers for wild type and Δphz cells equal? In other words, the data in this figure are normalized and presented as relative survival vs. the control group with no antibiotic. However, are the control groups equal? This would eliminate any concern that inoculum effect might be a contributor to perceived changes in antibiotic sensitivity. Please note, however, that if the cell numbers are not equivalent in wild type and mutant biofilms, that the interpretation would likely be called into question by many in the field.

The reviewer is correct--this figure only shows the normalized data--but Supplementary Figure 3 shows CFU counts for untreated biofilms and indicates that the control groups are equal. We tested survival for WT, Δ phz, and several other strains (including those relevant for Figures 2 and 4) and found that growth in the absence of antibiotic (i.e. the data used for normalization) did not show any significant differences. To make this more clear for the reader, we now reference Supplementary Figure 3 in the legend for Figure 1 and also refer to it in the main text (lines 89-91).

- Figure 2. Could the authors please provide the letter designations for the pel and mex genes that were deleted for this analysis? In the case of the engineered mex mutant bearing multiple mutations, this information might be better in the figure legend.

We have added the letter designations to the figure legend.

- Figure 4. Could the authors please maintain the same y-axis scale (i.e. range) in panels b and d to enable easier comparisons for the reader? Is it the same data in panel b as in panel d for the wild type group (it looks conspicuously similar)? Why not combine data in b and d into a single panel? Lastly, same criticism as for Figure 1 above – are there equivalent starting numbers of cells in wild type and mutant biofilms?

Because we were unable to generate the IdhA complementation strain for Δ phz Δ IdhA, we have decided to remove panel b from the paper. The data for the biofilms not treated with antibiotics is shown in Supplementary Figure 3 and we detected no significant differences between the strains in the absence of ciprofloxacin. We added a reference to Supplementary Figure 3 to the legend for Figure 4.

- Figure S4. It might be simpler to present this information in a line graph in a single figure panel. Was there a reason to split this information across four panels?

We agree and combined the data into a single line graph (this is now Supplementary Figure 5 in the updated manuscript).

Reviewer #3 (Remarks to the Author):

Review comments on

Ref.: NCOMMS-18-23186

Title: Phenazine production promotes antibiotic tolerance and metabolic heterogeneity in *Pseudomonas aeruginosa* biofilms

by Prof Dietrich and co-worker

This manuscript described the application of microsensors and stimulated Raman scattering (SRS) microscopy to detect redox profiles and metabolic activity of cross-sectioned colonies. The author regarded colonies as biofilms, however, it is hard to equal colonies to biofilms. Actually it

has been shown that colonies of *Pseudomonas aeruginosa* are more similar to planktonic cells instead of biofilm in terms of protein profiles (Mikkelsen et al., J Bacteriol (2007) 189:2411-2416). Although the work used a few techniques to measure the cross-section of the colonies, the findings are not robust and it is unclear why the cross-sectioned colonies are relevant to biofilm, given that colonies are not really biofilm. This manuscript may be more suitable to a specific journal.

A biofilm, by definition, is a consortium of microbial cells encased in a self-produced matrix. Hallmarks of biofilms are the formation of nutrient gradients, heterogeneity in gene expression and increased antibiotic resistance as compared to liquid cultures. Bacterial colonies exhibit all of these characteristics and have been used to investigate aspects of biofilm physiology for years by numerous research groups working with a diversity of microbes (references provided below). In our study, we find that colonies are ~100-1000x more resistant to antibiotics compared to stationary-phase liquid cultures. One of our major findings is that phenazines promote the antibiotic resistance of colony biofilms but not that of stationary-phase cultures, highlighting the physiological differences between these two modes of growth.

Mikkelsen et al. reported that, under their experimental conditions, the proteomes of flow-cell biofilms were similar to those of exponential-phase liquid cultures, while the proteomes of colonies were more similar to stationary-phase liquid cultures (which they refer to as “planktonic”). The authors also stress that reported expression measurements dramatically differ presumably due to differences in experimental conditions, which calls the use of global expression patterns as a diagnostic tool for “biofilms” into question. Instead, “biofilm” may better be used as a general term for matrix-encased bacterial aggregates.

References:

Vlamakis H, Aguilar C, Losick R, Kolter R. Genes Dev. 2008. 22(7):945-53.

Fong JC, et al. Elife. 2017. 1;6. pii: e26163.

Sarenko O, et al. MBio. 2017. 8(5). pii: e01639-17.

Kolter R, Greenberg EP. 2006. Nature. 441:300–2.

Main concerns:

1. The deuterium from D₂O or deuterated D₇-glucose is able to integrate into cells within 20 minutes (Reference 35, Berry et al., PNAS 2015). It is not surprising that fresh cells on the surface of a colony lost deuterium after subsequently growing in 20mM unlabelled glucose for 12 hours (Fig. 5). What is the diffusion effect of D₂O and ciprofloxacin (Fig. 3)? What is the control result without D₂O?

In the colony biofilm system used here, nutrients as well as D₂O are supplied from the agar interface, i.e. the bottom of the biofilm. Nevertheless, we observed strong deuterium incorporation in the upper region of the biofilm, but little incorporation in lower regions (see Figure 3b), suggesting that D₂O and glucose can diffuse throughout the biofilm. The strong deuterium signal in the upper region is therefore determined by the metabolic activity of the cells,

not by the diffusion of D₂O and glucose (which would lead to a maximal signal in the lower region of the biofilm).

As suggested by the reviewer, we now present results for control biofilms, which were not exposed to D₂O (described in lines 165-166). These are included as the new Supplementary Figure 6, which shows negligible signal and background-free detection in the SRS images.

2. One of important conclusion is that phenazines promote metabolic activity of cells in microaerobic biofilm. Without the distribution data of phenazines, it is hard to link phenazine production and ciprofloxacin effect across the section.

The redox profile shown in Figure 3a indicates that phenazine reduction occurs across biofilm depth. We have also added data for a reporter strain in which fluorescent protein expression is driven by the transcription factor SoxR, which specifically senses oxidized phenazines (Supplementary Figure 9d). We obtain a strong fluorescence signal in the hypoxic/anoxic zone, indicating oxidized phenazines enter cells in this region. The co-localization of phenazine redox-cycling and deuterium signal suggests that phenazines support metabolism in the microaerobic region. These results are also now described in the main text (lines 191-196).

3. SRS is an important technique in the manuscript. Can the authors provide the information how SRS setting is used to detect C-D band. For example, the resolution of SRS spectra. What are original SRS spectra and statistical analysis?

We apologize for not providing sufficient detail in our explanation of SRS microscopy as applied in this study. We have now provided more information on SRS imaging of carbon-deuterium bonds (C-D) in the text (lines 165-166; lines 412-417) and specified that the spectral resolution of SRS imaging is FWHM = 6-7 cm⁻¹ in the method section (lines 422-423). By matching the frequency difference of pump (864 nm) and Stokes (1064 nm) lasers in SRS with that of C-D bond vibration (~2175 cm⁻¹), the Raman scattering cross section of C-D bonds can be greatly enhanced through stimulated emission. Sensitive detection is achieved by measuring the intensity loss in the pump laser with a lock-in amplifier.

We have also added the spontaneous Raman spectra of biofilms with deuterium labeling in the new Supplementary Figure 6, which show clear C-D peaks in the cell spectral-silent region. It is known that SRS has high spectral fidelity without distortion and identical spectra as in the spontaneous Raman measurement (Min et al. Annu Rev Phys Chem. 2011. 62: 507–530).

4. Many data have no replicates and error bars (Fig. 3a, 3b and 3c, Fig. 4c).

Figure 3a includes error bars already; for the oxygen profile, they are very small and might have been missed by the reviewer. The number of replicates and the nature of the error bars are indicated in the legend.

For clarity, we showed representative data in Figures 3b and 3c and provided the corresponding replicates and error bars in Supplementary Figure 8. In the manuscript, we only included the replicates for the paraffin thin sections (Figure 3b, top), since these are the focus of the study

and yield a higher signal-to-noise ratio than optical sections, which are imaged through the colony (Figure 3b, bottom). We show a plot of all replicates of the optical sections below. Pre-grown colony biofilms were transferred to medium containing 50% D₂O for a 12-hour period before imaging. The distribution of deuterium signal in optical sections in WT (a) and Δphz (b) for all 3 replicates (indicated in different colors) is shown. In this labeling regime, metabolic activity is correlated with an increase in deuterium signal. Deuterium signal in data plots is corrected for light scattering using the protein channel and normalized to the maximum signal.

For Figure 4c, we have now included a presentation of all replicates as Supplementary Figure 9e.

We now better highlight this by referring to Supplementary Figures 8 and 9 in the legends for figures 3 and 4 and in the main text.

Other comments

1. Fig. 3a, the finest tip of oxygen microsensor is 8-12 μm (<http://www.unisense.com/O2/>), how can the author produce oxygen profile with 5 μm resolution?

We use a default step size of 5 μm for microsensor probing, but do not claim that this produces data with 5- μm resolution. Nevertheless, for reference, we have provided profiles taken along the z-axis at 5- μm and 20- μm step sizes below. Please note that the slope of oxygen concentrations along the depth of the biofilm is not affected by the step size.

2. Figure 3b, how many replicates have been done and what is the errors? What is the controls?

As described in our response to point 4 above, the replicates for the paraffin thin section data of Figure 3b are shown in Supplementary Figure 8a. Five to six replicates were performed with reproducible results. This information is now also provided in the figure legend. For the optical section data, we have three replicates shown in response to comment 1 above.

We appreciate the reviewer pointing out the lack of control data. We now include a sample that was not labelled with deuterium, which shows negligible signal, in the new Supplementary Figure 6.

3. What is the diffusion effect of phenazine when it is added externally. What is the diffusion effect of medium, glucose and D2O?

We have addressed these questions in our response to main concerns (1) and (2) above. Because biofilms are mostly aqueous, diffusion coefficients for water are typically applied to biofilms, as exemplified by the references below.

Stewart, P. S. et al. Reaction-diffusion theory explains hypoxia and heterogeneous growth within microbial biofilms associated with chronic infections. NPJ Biofilms Microbiomes 2, 16012 (2016).

Stewart, P. S. Diffusion in biofilms. J. Bacteriol. 185 (5), 1485–1491 (2003).

REVIEWERS' COMMENTS:

Reviewer #4 (Remarks to the Author):

The authors have thoroughly addressed all issues raised by reviewer 3. They included requested additional information and experiments in the revised manuscript as well as in Supplementary section.